# Making Multimodal Generation Easier: When Diffusion Models Meet LLMs

## Abstract

We present EasyGen, an efficient model designed to enhance multimodal under-standing and generation by harnessing the capabilities of diffusion models and large language models (LLMs). Unlike existing multimodal models that predominately depend on encoders like CLIP or ImageBind and need ample amounts of training data to bridge the gap between modalities, EasyGen is built upon a bidirectional conditional diffusion model named BiDiffuser, which promotes more efficient interactions between modalities. EasyGen handles image-to-text generation by integrating BiDiffuser and an LLM via a simple projection layer. Unlike most existing multimodal models that are limited to generating text responses, EasyGen can also facilitate text-to-image generation by leveraging the LLM to create textual descriptions, which can be interpreted by BiDiffuser to generate appropriate visual responses. Extensive quantitative and qualitative experiments demonstrate the effectiveness of EasyGen, whose training can be easily achieved in a lab setting.

## 1 Introduction

Recent times have been remarkable progress in the field of artificial intelligence generated content (AIGC), notably in technologies like large language models (LLMs) (Chiang et al., 2023; Touvron et al., 2023; Brown et al., 2020; Chowdhery et al., 2022; Zeng et al., 2022) for text generation and diffusion models Rombach et al. (2022); Nichol et al. (2022); Saharia et al. (2022) for visual generation. These breakthroughs have paved the way for the development of large-scale multimodal generative models, sparking a recent trend of incorporating extra visual modules into LLMs. Collaborative models, such as Visual ChatGPT (Wu et al., 2023a) and MM-REACT (Yang et al., 2023), strategically use externally pre-trained tools to translate visual information into text descriptions and feed the data into LLMs. However, they are exclusively dependent on pre-trained tools for inference. Contrarily, end-to-end trained models including the BLIP series (Li et al., 2023b), LLaVA (Liu et al., 2023), MiniGPT-4 (Zhu et al., 2023), and mPLUG-Owl (Ye et al., 2023) focus on mapping image information to the text space of LLMs, enabling LLMs to comprehend visual inputs.

Existing end-to-end models are also not without limitations. First, most of these multimodal models rely on either CLIP (Radford et al., 2021) or ImageBind (Girdhar et al., 2023) as their image encoder. While these encoders excel in learning unified representations that encompass both text and images, they face challenges when it comes to transforming between different modalities. This predicament makes current vision-language models relying heavily on sizable data sets to align CLIP/Bind-encoded images with the language model, due to the disparity between different modalities. Furthermore, most of existing multimodal models lack the ability of generating multimodal responses, as they are primarily designed to understand multimodal content and often fall short in generating content beyond text. A recent work Emu (Sun et al., 2023) takes a unified approach to generate visual or textual tokens, but it also heavily relies on vast quantities of training data.

In this work, we propose EasyGen, a model that facilitates multimodal generation by harnessing the strengths of diffusion models and LLMs. Diffusion models with multiple skills, such as Versatile Diffusion (Xu et al., 2023) and UniDiffuser (Bao et al., 2023b), have exhibited remarkable capabilities in accurately capturing multimodal distributions. UniDiffuser, in particular, shows great potential in accommodating multimodal interactions. It views both image and text as sequential token streams for diffusion calculations, making it well-suited for the text representations in LLMs that are typically structured as sequences. However, since UniDiffuser aims to fit all conditional distributions (including

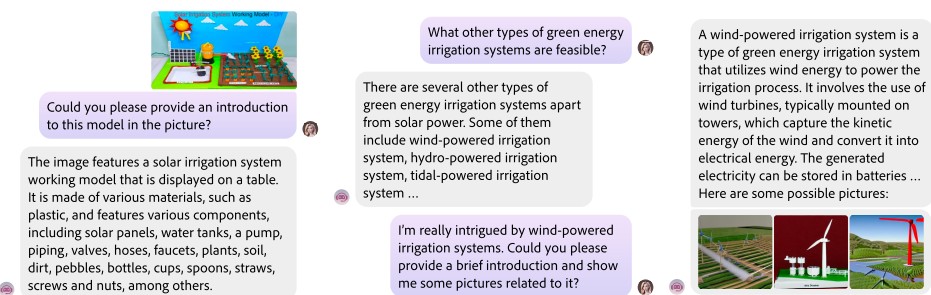

Figure 1: Our model EasyGen can understand multimodal inputs and generate multimodal responses, as illustrated by model-generated speech bubbles in grey color, which include both text and images.

those conditioned on noisy inputs) into one model, it is less effective on particular tasks such as conditional generation based on noise-free inputs. To address this limitation, we finetune UniDiffuser with a specific focus on the targeted image-to-text and text-to-image tasks. The finetuned model, referred to as BiDiffuser, forms a core component of EasyGen for text and image generation.

BiDiffuser is able to convert image data into a textual format, which simplifies the process of synchronizing its embedding space with that of an LLM for semantic comprehension and reasoning. As illustrated in Figure 2, we bridge BiDiffuser and the LLM using a simple projection layer, which can be trained efficiently with a small amount of data for image-to-text tasks such as image captioning and visual question answering. Alternatively, the LLM can be utilized to generate detailed descriptions and cues derived from text contexts like dialogues, which can aid BiDiffuser in generating accurate visual responses, as illustrated in Figure 2.

Figure 1 demonstrates the capability of EasyGen in processing multimodal inputs and generating the appropriate multimodal responses (see more examples provided in Appendix. H). Furthermore, EasyGen achieves competitive performance compared to state-of-the-art models with much less training data. It is worth noting that the training of EasyGen can be performed in a laboratory-level environment. Without employing parameter-efficient fine-tuning techniques like LoRa (Hu et al., 2021), EasyGen only requires about 120 A100 (80G) GPU hours during the pre-training process (for training BiDiffuser) and 20/72 A100 (80G) GPU hours during the alignment process for fine-tuning FlanT5XL/Vicuna-7B. By using LoRa, the training process of EasyGen can be significantly more efficient. For instance, the fine-tuning time for Vicuna-7B can be reduced from 70 to just 13 GPU hours (see Table 10).



Figure 2: Overview of EasyGen.

## 2  BASICS OF DIFFUSION MODELS

**Unconditional Generation**   Given a data sample taken from a real data distribution $\mathbf{x}_0 \sim q(\mathbf{x}_0)$, diffusion models (Sohl-Dickstein et al., 2015; Ho et al., 2020) first destruct the data by constructing a Markov forward process and gradually injecting noise to the data:

$$q(\mathbf{x}_{1:T}|\mathbf{x}_0) = \prod_{t=1}^{T} q(\mathbf{x}_t|\mathbf{x}_{t-1}), \quad q(\mathbf{x}_t|\mathbf{x}_{t-1}) = \mathcal{N}(\mathbf{x}_t; \sqrt{1-\beta_t}\mathbf{x}_{t-1}, \beta_t\mathbf{I}), \tag{1}$$

where $\beta_t \in (0, 1)$ is the variance added at diffusion step $t$. Then, they learn to reverse the process:

$$p(\mathbf{x}_{0:T}) = p(\mathbf{x}_T)\prod_{t=1}^{T} p_\theta(\mathbf{x}_{t-1}|\mathbf{x}_t), \quad p_\theta(\mathbf{x}_{t-1}|\mathbf{x}_t) = \mathcal{N}(\mathbf{x}_{t-1}; \mu_t(\mathbf{x}_t, t), \sigma_t^2\mathbf{I}), \tag{2}$$

where $p(\mathbf{x}_T) = \mathcal{N}(\mathbf{x}_T; 0, \mathbf{I})$ is the standard Gaussian distribution and $\mu_t(\cdot)$ is the parameterization of the predicted mean. Diffusion models are trained to maximize the marginal likelihood of the data

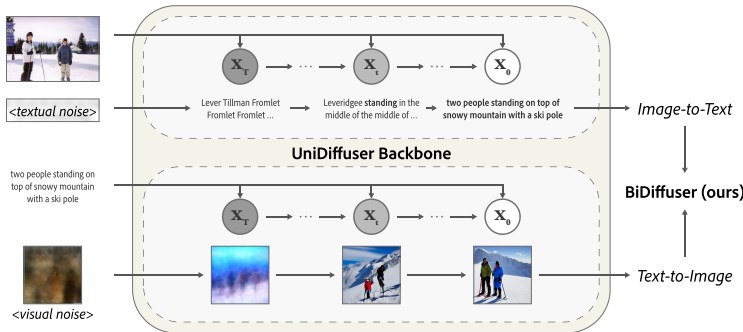

Figure 3: The training process of BiDiffuser involves finetuning UniDiffuser (Bao et al., 2023b) with a joint objective of image-to-text and text-to-image tasks.

$\mathbb{E}[\log p_\theta(\mathbf{x}_0)]$, and the canonical objective is the variational lower bound of $\log p_\theta(\mathbf{x}_0)$. Denoising diffusion probabilistic models (Ho et al., 2020) generate samples $\mathbf{x}_t \sim q(\mathbf{x}_t|\mathbf{x}_0)$ by injecting noise $\boldsymbol{\epsilon} \sim \mathcal{N}(0, \mathbf{I})$ to the data $\mathbf{x}_0$, and train a network $\boldsymbol{\epsilon}_\theta(\cdot)$ to predict the added noise $\boldsymbol{\epsilon}$ using a standard mean squared error loss:

$$\mathcal{L} := \mathbb{E}_{\mathbf{x}_0, \boldsymbol{\epsilon}, t}[\|\boldsymbol{\epsilon} - \boldsymbol{\epsilon}_\theta(\mathbf{x}_t, t)\|^2]. \tag{3}$$

Note that $\mu_t(\mathbf{x}_t, t)$ can be derived from $\boldsymbol{\epsilon}_\theta(\mathbf{x}_t, t)$.

**Conditional Generation**   For conditional generation, a paired data $(\mathbf{x}_0, \mathbf{y}_0)$ is given, and the aim is to model the conditional data distribution $q(\mathbf{x}_0|\mathbf{y}_0)$, where $\mathbf{y}_0$ can be image class or text prompt. Conditional generation includes classifier guidance (Dhariwal & Nichol, 2021) and classifier-free guidance (Ho & Salimans, 2021). Classifier guidance requires training an extra classifier on noisy data at inference time to improve the sample quality. For classifier-free guidance, no classifier needs to be trained. The denosing network $\boldsymbol{\epsilon}_\theta(\mathbf{x}_t|\mathbf{y}_0)$ simply conditions on the information encoded in $\mathbf{y}_0$. At inference time, with a guidance scale $s$, the modified score estimate is further in the direction of $\boldsymbol{\epsilon}_\theta(\mathbf{x}_t|\mathbf{y}_0)$ and away from the unconditional model $\boldsymbol{\epsilon}_\theta(\mathbf{x}_t|\emptyset)$ ($\emptyset$ is a null token) as follows:

$$\hat{\boldsymbol{\epsilon}}_\theta(\mathbf{x}_t|\mathbf{y}_0) = \boldsymbol{\epsilon}_\theta(\mathbf{x}_t|\emptyset) + s \cdot (\boldsymbol{\epsilon}_\theta(\mathbf{x}_t|\mathbf{y}_0) - \boldsymbol{\epsilon}_\theta(\mathbf{x}_t|\emptyset)). \tag{4}$$

## 3   EASYGEN: EASY MULTIMODAL GENERATION WITH A BIDIRECTIONAL CONDITIONAL DIFFUSION MODEL AND LLMS

We propose EasyGen, a model capable of processing multimodal inputs and generating multimodal outputs. It achieves easy multimodal generation by leveraging a bidirectional conditional diffusion model to effectively bridge the gap between different modalities and an LLM to comprehend multimodal tasks and produce textual responses containing cues for multimodal message creation. In the subsequent section, we outline the multimodal generation process of EasyGen.

### 3.1   BIDIFFUSER: A BIDIRECTIONAL CONDITIONAL DIFFUSION MODEL

Since the text space of LLMs is discrete, to minimize the disparity between the output of a diffusion model and the input of LLMs, we leverage Unidiffuser (Bao et al., 2023b), a unified diffusion model capable of transforming images into the discrete text space.

During the training process, UniDiffuser injects noise $\boldsymbol{\epsilon}^x$ and $\boldsymbol{\epsilon}^y$ to a set of paired image-text data $(\mathbf{x}_0, \mathbf{y}_0)$ and generates noisy data $\mathbf{x}_{t^x}$ and $\mathbf{y}_{t^y}$, where $0 \leqslant t^x, t^y \leqslant T$ represent two individual timesteps (perturbation levels). It then trains a joint noise prediction network $\boldsymbol{\epsilon}_\theta(\mathbf{x}_{t^x}, \mathbf{y}_{t^y}, t^x, t^y)$ to predict the noise $\boldsymbol{\epsilon}^x$ and $\boldsymbol{\epsilon}^y$ by minimizing the mean squared error loss:

$$\mathbb{E}_{\boldsymbol{\epsilon}^x, \boldsymbol{\epsilon}^y, \mathbf{x}_0, \mathbf{y}_0}[\|[\boldsymbol{\epsilon}^x, \boldsymbol{\epsilon}^y] - \boldsymbol{\epsilon}_\theta(\mathbf{x}_{t^x}, \mathbf{y}_{t^y}, t^x, t^y)\|^2], \tag{5}$$

where the output of $\boldsymbol{\epsilon}_\theta$ is the concatenation of the estimated noise $\boldsymbol{\epsilon}_\theta^x$ and $\boldsymbol{\epsilon}_\theta^y$, i.e., $\boldsymbol{\epsilon}_\theta = [\boldsymbol{\epsilon}_\theta^x, \boldsymbol{\epsilon}_\theta^y]$.

By predicting $\boldsymbol{\epsilon}_\theta(\mathbf{x}_{t^x}, \mathbf{y}_{t^y}, t^x, t^y)$ for any $t^x$ and $t^y$, UniDiffuser learns all distributions related to $(\mathbf{x}_0, \mathbf{y}_0)$ simultaneously. This includes all conditional distributions: $q(\mathbf{x}_0|\mathbf{y}_0)$ for text-to-image generation, $q(\mathbf{y}_0|\mathbf{x}_0)$ for image-to-text generation, and those conditioned on noisy input, i.e., $q(\mathbf{x}_0|\mathbf{y}_{t^y})$

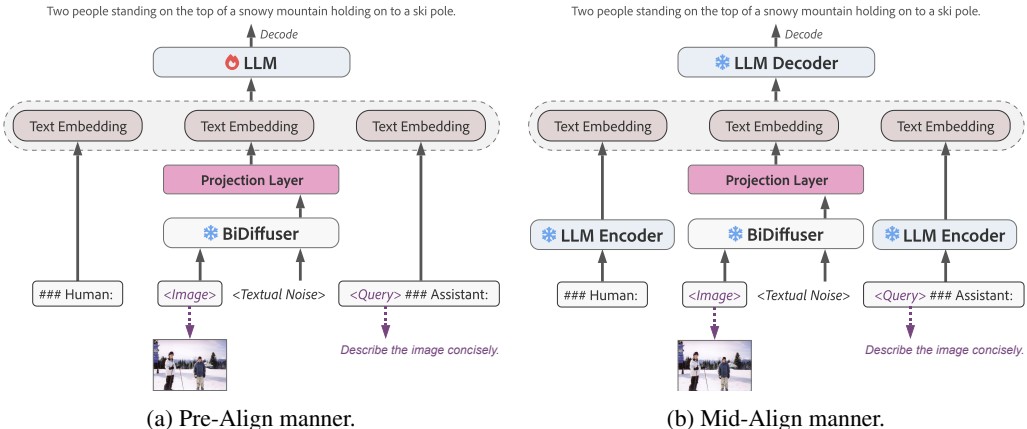

(a) Pre-Align manner.  (b) Mid-Align manner.

Figure 4: Two different ways of aligning BiDiffuser with LLMs.

and $q(\mathbf{y}_0|\mathbf{x}_{t^x})$, for $0 < t^x, t^y \le T$. Learning a conditional distribution $q(\mathbf{x}_0|\mathbf{y}_{t^y})$ or $q(\mathbf{y}_0|\mathbf{x}_{t^x})$ can be seen as learning a distinct task. From a multitask learning perspective, due to limited network capacity, learning many tasks simultaneously (i.e., fitting all distributions to a single network) may result in *task competition or task conflict*, ultimately leading to suboptimal performance in particular tasks such as $q(\mathbf{x}_0|\mathbf{y}_0)$ and $q(\mathbf{y}_0|\mathbf{x}_0)$.

To resolve this issue and enhance the performance of both image-to-text and text-to-image generation tasks, we finetune UniDiffuser with exclusive emphasis on the two tasks:

$$\mathcal{L} = \mathbb{E}_{\boldsymbol{\epsilon}^x, \boldsymbol{\epsilon}^y, \mathbf{x}_0, \mathbf{y}_0}[\|\boldsymbol{\epsilon}^x - \boldsymbol{\epsilon}_\theta^x(\mathbf{x}_{t^x}, \mathbf{y}_0, t^x, 0)\|^2 + \alpha\|\boldsymbol{\epsilon}^y - \boldsymbol{\epsilon}_\theta^y(\mathbf{x}_0, \mathbf{y}_{t^y}, 0, t^y)\|^2], \tag{6}$$

where $\alpha$ is a hyperparameter to balance the learning paces of the two tasks. As depicted in Figure 3, our training objective entails predicting the text $\mathbf{y}_0$ based on the input image $\mathbf{x}_0$ and vice versa, where the input conditions for the model are noise-free. We employ classifier-free guidance. During training, we estimate the noise injected to the image (i.e., $\boldsymbol{\epsilon}_\theta^x(\mathbf{x}_{t^x}, \mathbf{y}_0, t^x, 0)$) conditioned on the noise-free text $\mathbf{y}_0$ and the noise to the text (i.e., $\boldsymbol{\epsilon}_\theta^y(\mathbf{x}_0, \mathbf{y}_{t^y}, 0, t^y)$) given the noise-free image $\mathbf{x}_0$. During inference, with a guidance scale $s \geqslant 0$, we use the modified prediction $\hat{\boldsymbol{\epsilon}}_\theta$ to guide towards the condition:

$$\begin{aligned}
\hat{\boldsymbol{\epsilon}}_\theta^x(\mathbf{x}_{t^x}, \mathbf{y}_0, t^x, 0) &= \boldsymbol{\epsilon}_\theta^x(\mathbf{x}_{t^x}, \boldsymbol{\epsilon}^y, t^x, T) + s \cdot (\boldsymbol{\epsilon}_\theta^x(\mathbf{x}_{t^x}, \mathbf{y}_0, t^x, 0) - \boldsymbol{\epsilon}_\theta^x(\mathbf{x}_{t^x}, \boldsymbol{\epsilon}^y, t^x, T)), \\
\hat{\boldsymbol{\epsilon}}_\theta^y(\mathbf{x}_0, \mathbf{y}_{t^y}, 0, t^y) &= \boldsymbol{\epsilon}_\theta^y(\boldsymbol{\epsilon}^x, \mathbf{y}_{t^y}, T, t^y) + s \cdot (\boldsymbol{\epsilon}_\theta^y(\mathbf{x}_0, \mathbf{y}_{t^y}, 0, t^y) - \boldsymbol{\epsilon}_\theta^y(\boldsymbol{\epsilon}^x, \mathbf{y}_{t^y}, T, t^y)),
\end{aligned} \tag{7}$$

where $\boldsymbol{\epsilon}_\theta^x(\mathbf{x}_{t^x}, \boldsymbol{\epsilon}^y, t^x, T)$ ( $t^y = T$ and $\mathbf{y}_T = \boldsymbol{\epsilon}^y$) and $\boldsymbol{\epsilon}_\theta^y(\boldsymbol{\epsilon}^x, \mathbf{y}_{t^y}, T, t^y)$ ($t^x = T$ and $\mathbf{x}_T = \boldsymbol{\epsilon}^x$) represent the unconditional models when $T$ is sufficiently large. We name the finetuned model "BiDiffuser", signifying its specialized ability in bidirectional conditional generation.

## 3.2 IMAGE-TO-TEXT GENERATION

BiDiffuser can convert images into vectors in the text space, facilitating alignment with the vector space of LLMs. In the following, we show how BiDiffuser can be integrated with LLMs to perform image-to-text generation tasks such as image captioning and visual question answering (VQA).

### 3.2.1 ALIGNING BIDIFFUSER WITH LLMS

We connect BiDiffuser and LLMs via a simple projection layer, which maps text embeddings obtained from the output of the diffusion model to the embedding space of LLMs. As shown in Figure 4, the alignment can take place either prior to the LLM (referred to as Pre-Align manner) or between its encoder and decoder components (referred to as Mid-Align manner).

**Pre-Align Manner**  As shown in Figure 4a, the projection layer is placed before the LLM to map the output of BiDiffuser (image representations) to the text embedding space of the LLM. The text embedding of the input image is then concatenated with the embeddings of the textual instructions and fed to the LLM for decoding. To synchronize the text space of BiDiffuser with that of the LLM,

we propose to use the image-grounded text generation (ITG) objective to drive the model to generate texts based on the input image by computing the auto-regressive loss:

$$\mathcal{L}_{\text{ITG}} = -\frac{1}{L} \sum_{l=1}^{L} \log p_\theta(w_l^g | w_{<l}^g, I, T_I), \tag{8}$$

where $w^g = (w_1^g, ..., w_L^g)$ represents the ground-truth caption of image $I$ with length $L$, $T_I$ is the text instruction, and $\theta$ denotes the model parameters, which include the parameters of the projection layer and the LLM.

**Mid-Align Manner**  As shown in Figure 4b, the projection layer is placed between the LLM's encoder and decoder, aiming to map the output of BiDiffuser to the embedding space of the text that is encoded by the LLM's encoder. Particularly, we argue that the output of BiDiffuser, once mapped by the projection layer and denoted as $\mathbf{d}_{\text{diff}}$, should align with the image caption that is encoded by the LLM's encoder, denoted as $\mathbf{d}_{\text{llm}}$. Therefore, to accurately learn the alignment between the image and text representations, in addition to the ITG loss in Eq. 8, we also employ an image-text distance minimization (ITDM) loss:

$$\mathcal{L}_{\text{ITDM}} = \frac{1}{N} \sum_{i=1}^{N} \|\mathbf{d}_{\text{diff}} - \mathbf{d}_{\text{llm}}\|_2^2, \quad \mathcal{L}_{\text{mid}} = \mathcal{L}_{\text{ITG}} + \mathcal{L}_{\text{ITM}}. \tag{9}$$

where $N$ is the batch size, and $\mathcal{L}_{\text{mid}}$ is the overall loss. In this manner, the model parameters $\theta$ only include the parameters of the projection layer.

After aligning BiDiffuser with LLMs, EasyGen gains the capability of zero-shot image-to-text generation, which includes tasks such as image captioning and VQA.

### 3.2.2 Instruction-Tuning LLMs to Process Multimodal Tasks

Before aligning BiDiffuser with an LLM, we perform instruction-tuning on the LLM to equip it with the capability of understanding multimodal tasks. We construct the instruction data as follows. With reference to fastchat[*], we designed different forms of instructions for different LLMs:

FlanT5: ###Human: <image></Img> + <random[query]>. ###Assistant: <answer>.
Vicuna: USER: <image></Img> + <random[query]>. Assistant: <answer>.

For the <image> placeholder, we substitute it with one of the captions associated with the image. Note that an image can have multiple captions that convey a similar meaning. For each image, we randomly choose one of its captions, which is then fixed to be used specifically for the <answer> placeholder. As for <random[query]>, we randomly select a query from a predefined set of text queries that prompt the description of the given image as outlined in Table 8.

To avoid overfitting to the captioning task and counter the model's inclination to generate excessively short outputs, we have devised specific instructions (blue texts in Table 8), which enable the LLM to produce concise responses when necessary. Furthermore, we incorporate an additional 80K instances of multimodal instruction data from LLaVA (Liu et al., 2023), which helps to preserve the LLM's capability to generate comprehensive and detailed responses.

Moreover, to equip the LLM with the capability to comprehend various multimodal tasks, we curate distinct instruction templates for different tasks, as outlined in Appendix F.

### 3.3 Text-to-Image Response Generation

Most of existing multimodal models, including the BLIP series (Li et al., 2022), LLaVa (Liu et al., 2023), and MiniGPT4 (Zhu et al., 2023), are unable to provide a multimodal response as they are primarily designed to generate only textual outputs. On the other hand, Emu (Sun et al., 2023) takes a unified approach to predict the subsequent visual or textual token in an auto-regressive manner, but it is heavily reliant on vast quantities of training data. Contrary to the limitations of these existing models, EasyGen, by leveraging the bidirectional generation capability of BiDiffuser and the inference capability of LLMs, can produce accurate and high-quality visual response with ease.

---

[*]https://github.com/lm-sys/FastChat/tree/main

To tackle multimodal response generation tasks such as PhotoChat (Zang et al., 2021), we adopt the approach used in Divter (Sun et al., 2021) (note that Divter cannot encode and process visual images). First, we finetune the LLM to generate detailed image captions based on dialogue context. Then, we employ BiDiffuser to create the corresponding images with the produced captions. Specifically, we replace the image featured in the dialogue with its corresponding descriptive caption, encapsulating it with task-specific tokens ,</Img> and constructing the following instruction templates:

USER: Dialog history + <image></Img> + Dialog history. Assistant: <response>.
USER: Dialog history. Assistant: <response> + <image></Img>.

Note that when <image> appear in the response, it represents the generated description of the image. Training with the instruction data enables our model to not only produce text responses but also perform image intent classification and generate image captions that BiDiffuser can interpret.

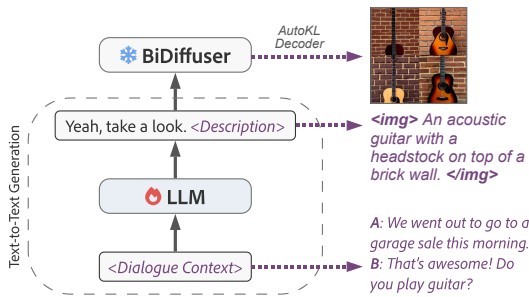

## 4 EXPERIMENTS

### 4.1 EXPERIMENTAL SETUP

We initialize the encoder-decoder LLM using pre-trained weights from FlanT5XL or decoder-only LLM from Vicuna-7B, along with the utilization of the diffusion module from BiDiffuser.

Figure 5: Text-to-image generation by EasyGen. (Bottom) LLM generates response and description of the image. (Top) BiDiffuser takes the description as input and generates images.

During the alignment process, we maintain the frozen state of the BiDiffuser. The statistics of the datasets for pre-training, alignment and instruction-tuning can be found in Appendix A. For the image captioning task, EasyGen is evaluated on both the MS-COCO (Lin et al., 2014) Karpathy test set and the NoCaps (Agrawal et al., 2019) validation set. For the VQA task, our method is evaluated on OK-VQA (Marino et al., 2019) validation set and GQA (Hudson & Manning, 2019) test-dev set.

To adapt the model for multimodal dialogue generation, we fine-tune the LLM and projection layer on the PhotoChat dataset. We incorporate photo-sharing activities into the dialogue context by generating <caption></Img>, and utilize cross-entropy loss exclusively for fine-tuning the multimodal generation task. Given the limited expressiveness of image descriptions in the PhotoChat dataset, as evidenced by Table 6's ground truth descriptions, we regenerate image annotations in a text format similar to that used in MS-COCO.

### 4.2 EVALUATION

We evaluate EasyGen on various vision-language tasks including image captioning (MS-COCO (Lin et al., 2014), NoCaps (Agrawal et al., 2019)), visual question answering (OK-VQA (Marino et al., 2019), GQA (Hudson & Manning, 2019)), and multimodal dialog generation (PhotoChat (Zang et al., 2021)). We use BLIP (Li et al., 2022), Flamingo (Alayrac et al., 2022), BLIP-2 (Li et al., 2023b), InstructBlip (Dai et al., 2023), MiniGPT-4 (Zhu et al., 2023), and LLaVA (Liu et al., 2023) as baselines for image-to-text tasks, and Maria (Liang et al., 2021) and Divter (Sun et al., 2021) as baselines for the multimodal response generation task. See details in Appendix B and Appendix C.

### 4.3 OVERALL RESULTS

Table 1 lists the automatic and ChatGPT evaluation results for each baseline and our models on MS-COCO and VQA datasets. EasyGen outperforms most of the baseline models on both the COCO test set and NoCaps validation set (zero-shot transfer). Although EasyGen is only pre-trained on a small dataset MS-COCO, its performance on the image captioning generation task is comparable to models (e.g., BLIP-2) pre-trained on a large dataset. This indicates that EasyGen can effectively combine the strength of diffusion module and LLM to generate smooth and informative captions. GPT scores do not vary significantly because the captions produced by the models in the image-captioning

| Model | Dataset Size | NoCaps (val) | | COCO (Karpathy) | | | OK-VQA | GQA |
|---|---|---|---|---|---|---|---|---|
| | | CIDEr | SPICE | BLEU@4 | CIDEr | GPT | Accuracy | Accuracy |
| BLIP (Li et al., 2022) | 129M | 113.2 | 14.8 | 40.4 | 136.7 | - | - | - |
| Flamingo (Alayrac et al., 2022) | 1.8B | - | - | - | 138.1 | - | 50.6 | - |
| BLIP-2 OPT-6.7B (Li et al., 2023b) | 129M | 121.0 | 15.3 | **43.5** | 145.2 | 8.4 | 36.4 | 36.4 |
| BLIP-2 FlanT5XL (Li et al., 2023b) | 129M | 121.6 | **15.8** | 42.4 | 144.5 | 8.3 | 39.4 | 44.4 |
| InstructBlip 7B (Dai et al., 2023) | 16M | **123.1** | - | 40.8 | 140.7 | - | 61.0* | 49.2* |
| MiniGPT-4 (Zhu et al., 2023) | 5M | 42.4 | - | - | - | - | 37.5 | 30.8 |
| LLaVA (Liu et al., 2023) | 753K | 33.1 | - | 7.9 | 30.0 | 8.6 | 54.4 | 41.3 |
| **EasyGen FlanT5XL** | 173K | 121.2 | 15.5 | **43.5** | **145.7** | 8.6 | 41.1 | 37.2 |
| **EasyGen Vicuna-7B** | 173K | 121.8 | **15.8** | 42.4 | 144.6 | **8.7** | 45.2 | **44.6** |

Table 1: Automatic evaluation and GPT evaluation of our model and the baselines on various vision-language tasks. The results of EasyGen on NoCaps, OK-VQA and GQA are obtained in a zero-shot setting. ⋆ indicates that the model was trained on other VQA datasets.

| Model | Response Generation | | | | Description Generation | | | Image |
|---|---|---|---|---|---|---|---|---|
| | BLEU-1 | BLEU-2 | PPL↓ | ROUGE-L | BLEU-1/2 | ROUGE-L | PPL↓ | FID↓ |
| Divter (Sun et al., 2021) | 6.5 | 1.7 | 59.63 | 5.69 | 15.1/11.4 | 15.81 | 5.12 | 29.16 |
| Maria (Liang et al., 2021) | 13.8 | 9.2 | 48.75 | 15.17 | - | - | - | - |
| **EasyGen FlanT5XL** | | | | | | | | |
| + w/ generated desc. | **22.3** | **18.7** | **4.32** | **17.24** | 13.5/10.2 | 13.84 | **4.16** | **10.30** |
| + w/o generated desc. | 17.8 | 12.4 | 7.61 | 15.12 | **17.4/13.2** | **16.71** | 6.23 | 75.46 |

Table 2: Automatic evaluation of our model and the baselines on the PhotoChat dataset.

task tend to be quite alike. For the OK-VQA and GQA dataset, the performance of EasyGen is improved compared with other models of a similar scale. For example, BLIP-2 adopts the task-special decoding method and achieves 39.4% accuracy on OK-VQA validation set, while ours can get 45.2% even with a simple decoding method, i.e., greedy search.

Table 2 lists the automatic evaluation results on the PhotoChat dataset. The results of Divter are cited from (Sun et al., 2021). We fine-tune Maria on PhotoChat dataset only for the response generation task. Since our EasyGen model can generate response and image description simultaneously, the response and description generation task has a similar PPL. Compared with other models, our method has clear advantages in the performance of PPL, indicating that by leveraging LLM, our model demonstrates strong performance on text generation tasks. Besides, we find that the image descriptions in the PhotoChat dataset are too concise to adequately convey the information of images. Therefore, we used the pre-trained model from the first stage to regenerate the image description (referred to as "w/ generated desc." in Table 2) which led to a large gap towards ground-truth descriptions, resulting in lower BLEU-1/2 and ROUGE-L. However, the performance of our model on BELU-1/2 and ROUGE is higher than other models on response generation tasks, indicating that introducing richer image descriptions is beneficial for generating more relevant and informative responses. We also provide some examples (Figure 6) to show the effectiveness of our method.

## 4.4 ABLATION STUDY

In Table 3, we investigate the impact of different training strategies on the model. After removing the ITDM loss, the performance of EasyGen is slightly weaker than the original model. It is evident that the MSE Loss can help to align the semantic spaces of the two models. Furthermore, the performance of the model will drop significantly after removing the cross-entropy loss, suggesting that constraints via the language model play a key role. Without the instruction tuning process on LLM, EasyGen has a significant decline in the performance of automatic evaluations, which indicates that prior tuning of the LLM to an accurate caption generation model is necessary.

In Table 4, we examine the impact of freezing/tuning BiDiffuser and the LLM. We conducted ablation studies on image captioning and VQA tasks. It can be observed that the frozen Mid-Align method outperforms the Pre-Align method in image captioning. This shows that the ITDM loss function is effective. However, the frozen Mid-Align method exhibits inferior performance in the VQA task.

| Model | NoCaps (val) | | COCO (Karpathy) | | | OK-VQA | GQA |
|---|---|---|---|---|---|---|---|
| | CIDEr | SPICE | SPICE | BLEU@4 | CIDEr | Accuracy | Accuracy |
| **EasyGen Mid-Align FlanT5XL** | 121.2 | **15.5** | **25.1** | **43.5** | **145.7** | 31.5 | 22.6 |
| + w/o ITDM | 118.6 | 15.3 | 24.8 | 42.2 | 141.5 | - | - |
| + w/o ITG | 93.2 | 12.9 | 23.0 | 35.1 | 127.6 | - | - |
| + w/o LLM pre-tuning | 110.8 | 14.5 | 24.4 | 40.7 | 139.6 | 25.8 | 18.1 |
| **EasyGen Vicuna-7B** | **121.8** | 15.3 | 24.9 | 42.4 | 144.6 | **45.2** | **44.6** |
| + w/o LLM pre-tuning | 107.3 | 14.3 | 24.2 | 40.1 | 137.5 | 44.1 | 41.2 |

Table 3: Ablation studies on the instruction-tuning process and loss functions.

| LLM | Diffusion Model | Alignment | NoCaps | COCO(Karpathy) | | | OK-VQA |
|---|---|---|---|---|---|---|---|
| | | | CIDEr | SPICE | BLEU@4 | CIDEr | Accuracy |
| ❄️ T5 | UniDiffuser | Pre-Align | 62.4 | 18.0 | 26.8 | 90.7 | 33.0 |
| 🔥 T5 | BiDiffuser | Pre-Align | 119.1 | **25.5** | 42.6 | 145.1 | **41.1** |
| ❄️ T5 | BiDiffuser | Mid-Align | 121.2 | 25.1 | 43.5 | **145.7** | 31.5 |
| 🔥 T5 | BiDiffuser | Mid-Align | 121.5 | 25.3 | **43.6** | **145.7** | 36.4 |
| 🔥 Vicuna-7B | BiDiffuser | Pre-Align | **121.8** | 24.9 | 42.4 | 144.6 | **45.2** |
| ❄️ Vicuna-7B | BiDiffuser | Pre-Align | 119.0 | 24.6 | 40.3 | 140.3 | 42.7 |

Table 4: Ablation studied on image captioning and VQA tasks. 🔥 / ❄️ represents we tune/freeze the weights of the LLM during the alignment process.

We hypothesize that this is due to the integration of mid-aligned target image features with query information, and the projection layer is insensitive to instruction information. We conduct instruction-tuning on Pre-Align T5 and Vicuna. Compared to models at the same scale, these instruction-tuned models achieve superior results. The results clearly demonstrate that the instruction tuned models outperformed other models significantly on the OK-VQA and GQA datasets.

## 4.5 FINE-TUNING EASYGEN FOR VQA TASKS

Considering the substantial cost involved in fine-tuning the diffusion model for VQA tasks, we opt to concatenate the output of BiDiffuser with the image encoded by image CLIP ViT-L/14 and fine-tune the parameters of the LLM and projection layers. We fine-tune the EasyGen on the training and validation splits from VQAv2, Text Captions, AOK-VQA and TextVQA datasets.

In order to verify the effectiveness of BiDiffuser, we also add this module to LLaVA Vicuna-7B and use the same mixture dataset to do instruction tuning. The details of training dataset can be found in Table 9. Since BiDiffuser can map images into text vectors, BiDiffuser can be directly migrated to the LLaVA Vicuan-7B model. We keep LLaVA and our model using the same instruction tuning datasets. Noting that LLaVA has used 595K pretraining data from CC-3M dataset (Sharma et al., 2018) to align CLIP and LLM. Our model does not pre-align CLIP and LLM, and only uses instruction-tuning data for training.

| Model | VQAv2 (test-dev) | MMbench (dev) | TextVQA |
|---|---|---|---|
| MiniGPT-4 (Zhu et al., 2023) | - | 24.3 | 19.4 |
| InstructBLIP Vicuna-7B (Dai et al., 2023) | - | 36.0 | 50.1 |
| LLaVA Vicuna-7B (Liu et al., 2023) | 77.6 | 43.6 | 44.1 |
| LLaVA Vicuna-7B + BiDiffuser | 78.2 | 45.7 | 46.7 |
| EasyGen ViT-L Vicuna-7B | 79.4 | 45.4 | 45.5 |
| + w/o BiDiffuser | 71.1 | 21.4 | 36.2 |

Table 5: Comparison with state-of-the-art open-ended generation models fine-tuned for visual question answering and benchmarks.

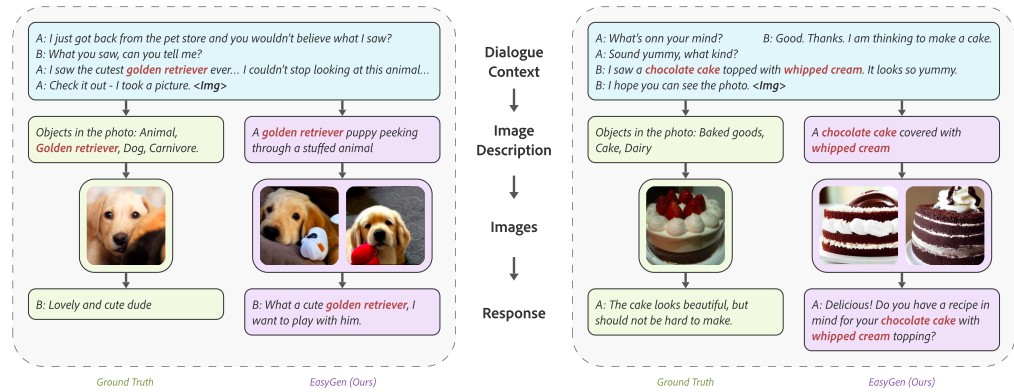

Figure 6: Examples of the generated responses on PhotoChat dataset. The text highlighted in red indicates the objects present in the image. The turns prefixed with A/B denote the given context.

## 5 RELATED WORK

**Multimodal Language Models**. Recent research has witnessed a surge of interest in multi-modal LLMs, including collaborative models such as Visual ChatGPT (Wu et al., 2023a), MM-REACT (Yang et al., 2023), and HuggingGPT (Shen et al., 2023), and end-to-end methods including Flamingo (Alayrac et al., 2022), Img2LLM (Guo et al., 2022), BLIP series (Li et al., 2023b; Dai et al., 2023; Li et al., 2022), BEiT series (Bao et al., 2021; Wang et al., 2022b), LLaVA (Liu et al., 2023), mPLUG-owl (Ye et al., 2023), MiniGPT-4 (Zhu et al., 2023), Llama-adapter (Zhang et al., 2023a), Otter (Li et al., 2023a), OFA (Wang et al., 2022a), and PaLI (Chen et al., 2022). In our works, EasyGen is built upon a bidirectional conditional diffusion model, which promotes more efficient interactions between modalities.

**Multimodal Diffusion Models**. Diffusion generative models (Rombach et al., 2022; Ramesh et al., 2021; Nichol et al., 2022; Ruiz et al., 2023) have achieved strong results in text conditioned image generation works. Specifically, Versatile Diffusion (Xu et al., 2023) employs the U-Net (Ronneberger et al., 2015) architecture with a multi-flow design to tackle multiple modalities and tasks, while UniDiffuser (Bao et al., 2023b) adopts the U-ViT (Bao et al., 2023a) framework to treat both image and text as sequential token streams for diffusion calculations. However, these models are unable to complete complex language tasks. EasyGen combines the advantages of diffusion models and LLMs and achieves competitive performance in both image-to-text and text-to-image tasks.

**Multimodal Response Generation**. Recent works have shown significant progress on multimodal response generation (Koh et al., 2023b; Aghajanyan et al., 2022; Zhang et al., 2023b; Wu et al., 2023b; Pan et al., 2023; Koh et al., 2023a). Divter (Sun et al., 2021) incorporates text-to-image generation into text-only dialogue response generation to produce a multimodal response. Leveraging the power of diffusion models, CoDi (Tang et al., 2023) can generate any combination of output modalities. Emu (Sun et al., 2023) takes a unified approach to predict the subsequent visual or textual token in an auto-regressive manne. In EasyGen, we efficiently combine the diffusion model and LLMs to generate multimodal outputs.

## 6 CONCLUSION

We have introduced EasyGen, a model that facilitates multimodal understanding and generation. In contrast to existing models that rely on encoders like CLIP or ImageBind (Girdhar et al., 2023) and require significant amounts of training data to integrate different modalities, EasyGen offers a more efficient solution by employing a bidirectional diffusion model named BiDiffuser. This allows for more effective modal interactions, handling both image-to-text and text-to-image generations by the fusion of BiDiffuser and LLMs. Comprehensive experiments underscores EasyGen's effectiveness and efficiency.

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

# A    DATASETS

We test the effectiveness of EasyGen by experimenting on different tasks including image captioning, visual question answering (VQA), and multimodal dialogue tasks.

We use the MS-COCO (Lin et al., 2014) dataset for image captioning. Following BLIP-2 (Li et al., 2023b), we fine-tune EasyGen on MS-COCO and evaluate its performance on the Karpathy test set and the NoCaps (Agrawal et al., 2019) validation set. In MS-COCO, each image typically has five captions that convey similar meanings. The training set consists of 82,783 images with 414,113 captions, while the COCO Karpathy test set has 5,000 images and the NoCaps validation set has 4,500 images.

For multimodal dialogue, we utilize the PhotoChat (Zang et al., 2021) dataset, which is a high-quality dataset consisting of 10,917 images and 12,286 dialogues. Each dialogue is associated with a user image and its corresponding text description. The dataset is divided into 10,286 training instances, 1,000 development instances, and 1,000 testing instances. Moreover, PhotoChat includes photo-sharing activities, defined as the process of creating <caption></Img> in this study. Each conversation in PhotoChat is broken down and constructed into multiple samples so that each round of responses can be learned. Specifically, we regard the first three turns as the dialog context, and the subsequent turns as the prediction targets. By converting the dialogues of this dataset into the form mentioned in 3.3, we obtained 49,240 train, 4,792 dev, and 4,836 test dialogue pairs.

For the VQA task, we conduct a quantitative evaluation on both the OK-VQA (Marino et al., 2019) validation set (5,046 questions) and the GQA (Hudson & Manning, 2019) test-dev set (12,578 questions). As shown in Table 4, for the frozen LLM, following BLIP-2, we employ the length penalty in beam search to encourage short answer generation. On the contrary, for the tuned LLM, we use the VQA instructions (as shown in Table 7) to do instruction tuning during the alignment process. The data used for instruction tuning is constructed by randomly selecting 5K data from the VQAv2 (Goyal et al., 2017) training set and 5K data from the Visual Dialog (Murahari et al., 2019) training set.

# B    BASELINES

We compare our proposed model with the following state-of-the-art baselines:

**BLIP** (Li et al., 2022) is a multimodal mixture of encoder-decoder. It can be used as an image-based text encoder or decoder. We use it to perform caption generation and VQA tasks.

**BLIP-2** (Li et al., 2023b) is pre-trained through bootstrapped learning from frozen visual encoder models and LLMs using an efficient pre-training strategy. We use it to perform caption generation and VQA tasks.

**Flamingo** (Alayrac et al., 2022) incorporates new cross-attention layers into Chinchilla language model (Hoffmann et al., 2022) to inject visual features, and pre-trains the new layers on billions of image-text pairs. We use it to perform caption generation and VQA tasks.

**InstructBlip** (Dai et al., 2023) is a vision-language instruction tuning framework that is trained with BLIP-2 and capable of solving various visual language tasks.

**MiniGPT-4** (Zhu et al., 2023) utilizes a single projection layer to align visual information from a pre-trained vision encoder with an LLM. Note that they employ the same visual encoder as used in BLIP-2.

**LLaVA** (Liu et al., 2023) employs a solitary projection layer to convert image features extracted from the pre-trained CLIP-ViT-L/14 visual encoder into the language embedding space of Vicuna (Chiang et al., 2023).

**Maria** (Liang et al., 2021) is a neural conversation agent which can leverage visual world experiences sourced from a vast image index. It possesses the ability to fetch a relevant image specific to the conversation and extract a wealth of visual knowledge from it.

**Divter** (Sun et al., 2021) focuses on exploring multimodal dialogue generative models. Given the dialogue context, this model first generates a text response or image description and then generates an image according to the description.

## C  IMPLEMENTATION DETAILS

**LLM**   During the alignment process, we utilize the AdamW optimizer with $\beta_0 = 0.9$, $\beta_1 = 0.99$, and weight decay of 0. The LLMs are trained with a cosine learning rate of 2e-5 and a warmup rate of 0.03. We use a batch size of 96 for the frozen LLMs and 32 for the tuned LLMs. During training, we convert the LLMs (FlanT5XL/Vicuna-7B) to BFloat16/FP16 and BiDiffuser to FP16.

**Diffusion Module**   We inherit the settings from UniDiffuser and utilize pre-trained weights from its checkpoint for our text-to-image generator. The model is fine-tuned on the MS-COCO dataset, which contains images with a resolution of $512 \times 512$, for 25K iterations with a batch size of 312. For all of our sampling processes, we employ DPM-Solver with 50 steps.

## D  LEARNING CURVES

As explained in Section 3.1 and Section 3.2.1, we perform fine-tuning of BiDiffuser on the MS-COCO dataset and instruction-tuning of the LLMs on the MS-COCO and LLaVA 80K datasets. Figure 7 displays the loss curves during the Mid-Align and Pre-Align training stages of FlanT5XL respectively. It can be seen that the utilization of BiDiffuser in the fine-tuning process exhibits a notable enhancement in performance across both stages, as compared to UniDiffuser.

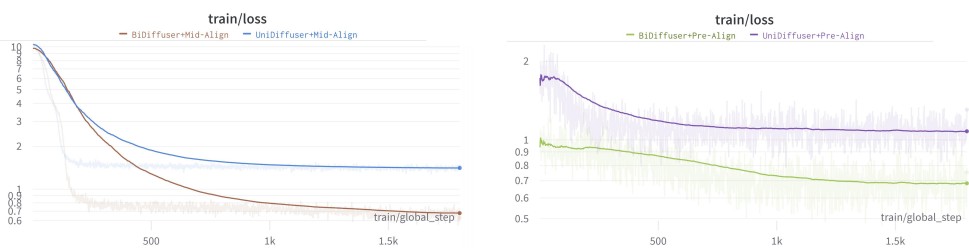

Figure 7: The learning curves of fine-tuning BiDiffuser/UniDiffuser for image captioning.

## E  EVALUATION

For evaluating the quality of text generation, we utilize metrics such as BLEU, Rouge-L, Accuracy, and PPL (Perplexity). Additionally, following the approach of Vicuna (Chiang et al., 2023) and LLaVA (Liu et al., 2023), we employ ChatGPT to assess the generated responses from our model. Specifically, for the image captioning task, we randomly select 30 images from the MS-COCO Karpathy split and then let ChatGPT score the responses generated by EasyGen and the baseline models. ChatGPT evaluates the models' responses based on relevance, details, and accuracy and assigns an overall score between 1 and 10, with a higher score indicating better performance.

To evaluate the quality of image generation, we use the Frechet Inception Distance (FID) score (Heusel et al., 2017), which measures the divergence between two multivariate normal distributions.

## F  INSTRUCTION TUNING

We list the instructions for different tasks in the main paper in Table 7. Specifically, the queries used to describe the image content are presented in Table 8. Table 7 shows the templates used in Vicuna, if the LLM employed is FlanT5, kindly use "Human" to substitute "USER" in the instruction templates.

| | Dataset | Task | Split | Metric |
|---|---|---|---|---|
| **Image-to-Text** | MS-COCO (Lin et al., 2014) | Image captioning | Test | CIDEr, BLEU, SPICE |
| | NoCaps (Agrawal et al., 2019) | Image captioning | Val | CIDEr, SPICE |
| | OK-VQA (Marino et al., 2019) | VQA | Val | Accuracy |
| | GQA (Hudson & Manning, 2019) | VQA | Test | Accuracy |
| **Multimodal Generation** | PhotoChat Zang et al., 2021 | Image dialogue | Test | PPL, BLEU, ROUGE, FID |

Table 6: Summary of the evaluation datasets and metrics.

| Task | Instruction Template |
|---|---|
| **Image Captioning** | USER: <image>+random[query] Assistant: |
| **LLaVA 80K** | USER: Please answer question from this image: <image> Question: <question> Assistant: 
 USER: Image: <image> Question: <question> Assistant: 
 USER: Answer question <question> through the image <image> Assistant: |
| **Multimodal Dialogue** | USER: Dialog history+<photo>+Dialogue history Assistant: |
| **VQA** | USER: Image: <image> Question: <question> Short answer: Assistant: |

Table 7: Examples of task instruction templates. <image> represents the input image, <question> denotes the question in the VQA and LLaVA 80K dataset, and <photo> is the image description of the input image.

1. Describe the image concisely.
2. Provide a brief description of the given image.
3. Can you describe this image briefly?
4. Provide a summary of the visual elements depicted in the image.
5. Give me the essential characteristics of the photograph in a concise manner.
6. Rephrase the image depicted in a concise manner.
7. Describe the objects in this image no in detail.
8. Please introduce the image for me briefly.
9. Give me the image's descriptions.
10. Please provide a general depiction of the image presented.

Table 8: For the image captioning task, a query instruction is randomly selected.

| Data types | Dataset | Size | BiDiffuser | Alignment | Fine-tuning |
|---|---|---|---|---|---|
| Caption | MS-COCO caption (Lin et al., 2014) | 83K | ✔ | ✔ | ✗ |
| | Visual Genome (Krishna et al., 2017) | 86K | ✔ | ✗ | ✗ |
| Multimodal instruction | LLaVA dataset Liu et al. (2023) | 80K | ✗ | ✔ | ✔ |
| VQA | VQAv2 (Goyal et al., 2017) | 83K | ✗ | - | ✔ |
| | AOK-VQA (Schwenk et al., 2022) | 66K | ✗ | ✗ | ✔ |
| OCR-related tasks | Text Captions (Sidorov et al., 2020) | 22K | ✗ | ✗ | ✔ |
| | TextVQA (Singh et al., 2019) | | ✗ | ✗ | ✔ |

Table 9: Description of datasets used in our alignment and VQA fine-tuning stages. Noting that in alignment process, we used 5K images from VQAv2 dataset.

| Model | Trainable Param. | Training Images | Training Cost |
|---|---|---|---|
| *Pre-training* | | | |
| BiDiffuser | 952M | 169K | 120 (A100 80GB) GPU hours |
| *Alignment* | | | |
| Projection Layer + ❄️ T5XL | 4M | 163K | 20 (RTX3090 24GB) GPU hours |
| Projection Layer + 🔥 T5XL | 3B | 173K | 20 (A100 80GB) GPU hours |
| Projection Layer + 🔥 Vicuna | 7B | 173K | 72 (A100 80GB) GPU hours |

Table 10: EasyGen's trainable parameters, training data size, and training cost during alignment process.

Table 9 shows the statistics of the pre-training datasets for BiDiffuser, alignment and VQA tasks. The VQA model is finetuned with the LM loss using ground-truth answers as targets. For finetuning, the input image resolution is set to 64 × 4096. We finetune the EasyGen model on mixture datasets for 1 epoch with a batch size of 32. We adopt the AdamW optimizer with $\beta = (0.9, 0.99)$ with the learning rate is 2e-5. We use a cosine learning rate decay with a learning rate is 2e-5 and warmup ration is 0.03.

## G  TRAINING EFFICIENCY

Table 10 summarizes the key factors in training EasyGen. The training process of EasyGen is computationally efficient, especially with the utilization of the parameter-efficient fine-tuning method LoRa (Hu et al., 2021). To enable multimodal response generation, we further train the aligned EasyGen. This process entails fine-tuning the LLM (FlanT5XL) on the PhotoChat dataset for 3 epochs, which typically requires approximately 4 A100 (80G) GPU hours.

## H  MORE QUALITATIVE RESULTS

We present several instances of the image-captioning task in Figure 8. In Figure 9, Figure 11 and Figure 10, we compare EasyGen with state-of-the-art multimodal language models. The responses of MiniGPT-4, LLaVA, mPLUG-owl and InstructBlip are obtained from their official demos. Morever, in Figure 13 and Figure 12, we show EasyGen's ability to accept multimodal inputs and generate multimodal responses.

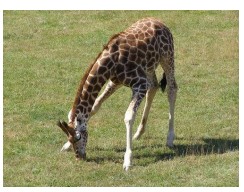

EasyGen: A giraffe eating grass on a green grass field.

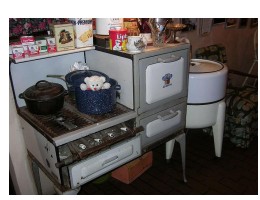

EasyGen: A kitchen with a stove and oven in a fireplace.

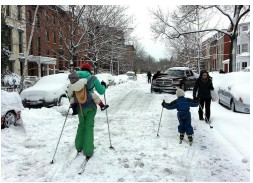

EasyGen: Two children are riding on skis with their parents in a snowy street.

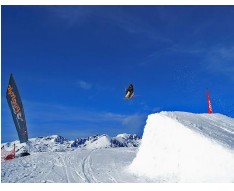

EasyGen: A snowboarder flying through the air on a frozen mountain slope.

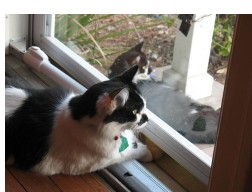

EasyGen: Two cats separated by the window are looking forward at the same time.

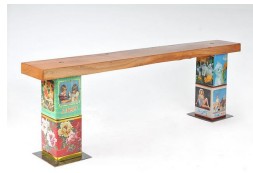

EasyGen: A long wooden bench with pictures of the world.

Figure 8: Examples of image captioning results by EasyGen.

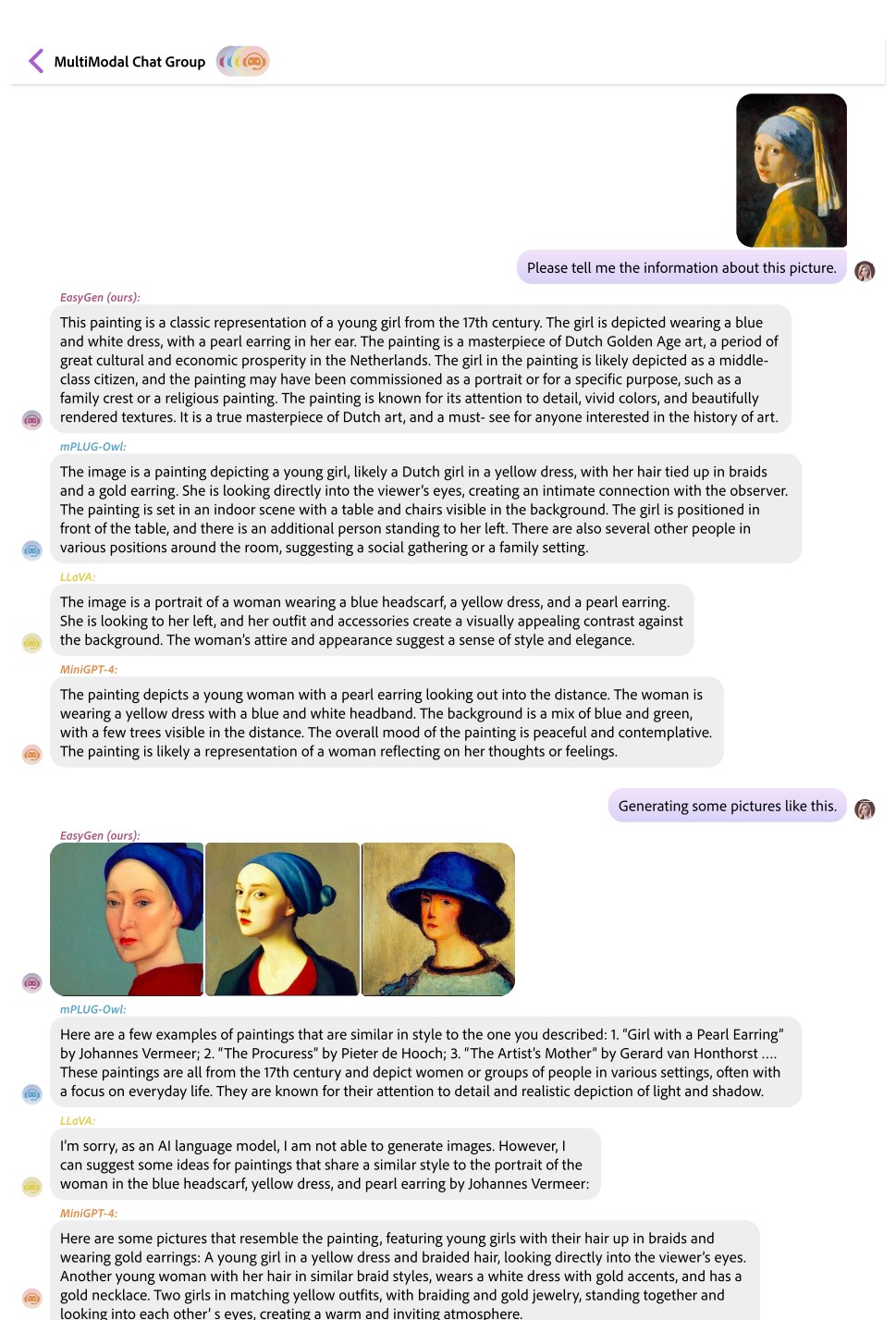

Figure 9: In this case study, for the first question, EasyGen can give an accurate answer including the background information of the image. With the image generation ability of BiDiffuser, EasyGen can generate visual responses.

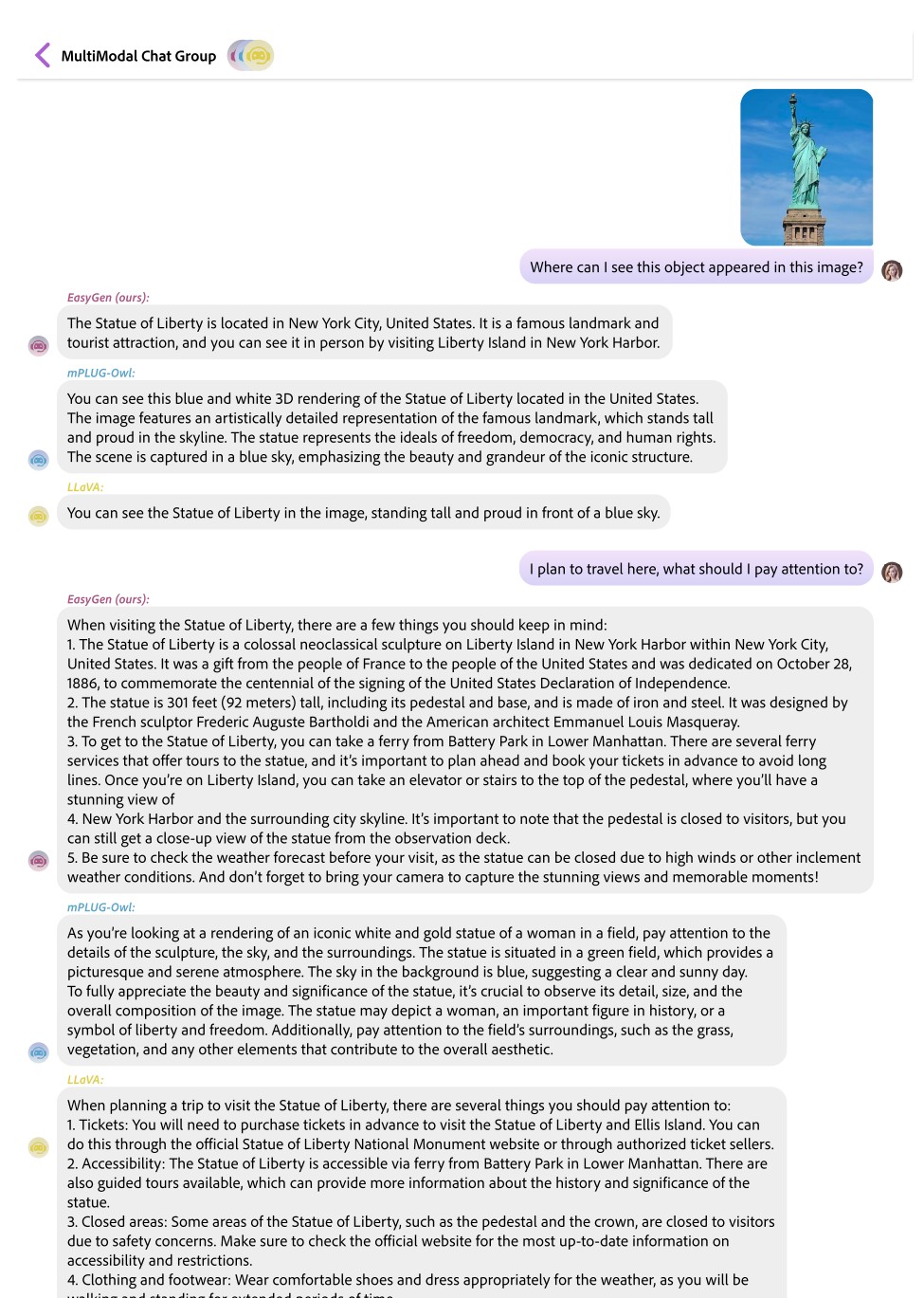

Figure 10: In this case study, for the first question, EasyGen can give an accurate answer, but the responses of the other two models are a bit biased. For the second question, EasyGen and LLaVA both give reasonable advice.

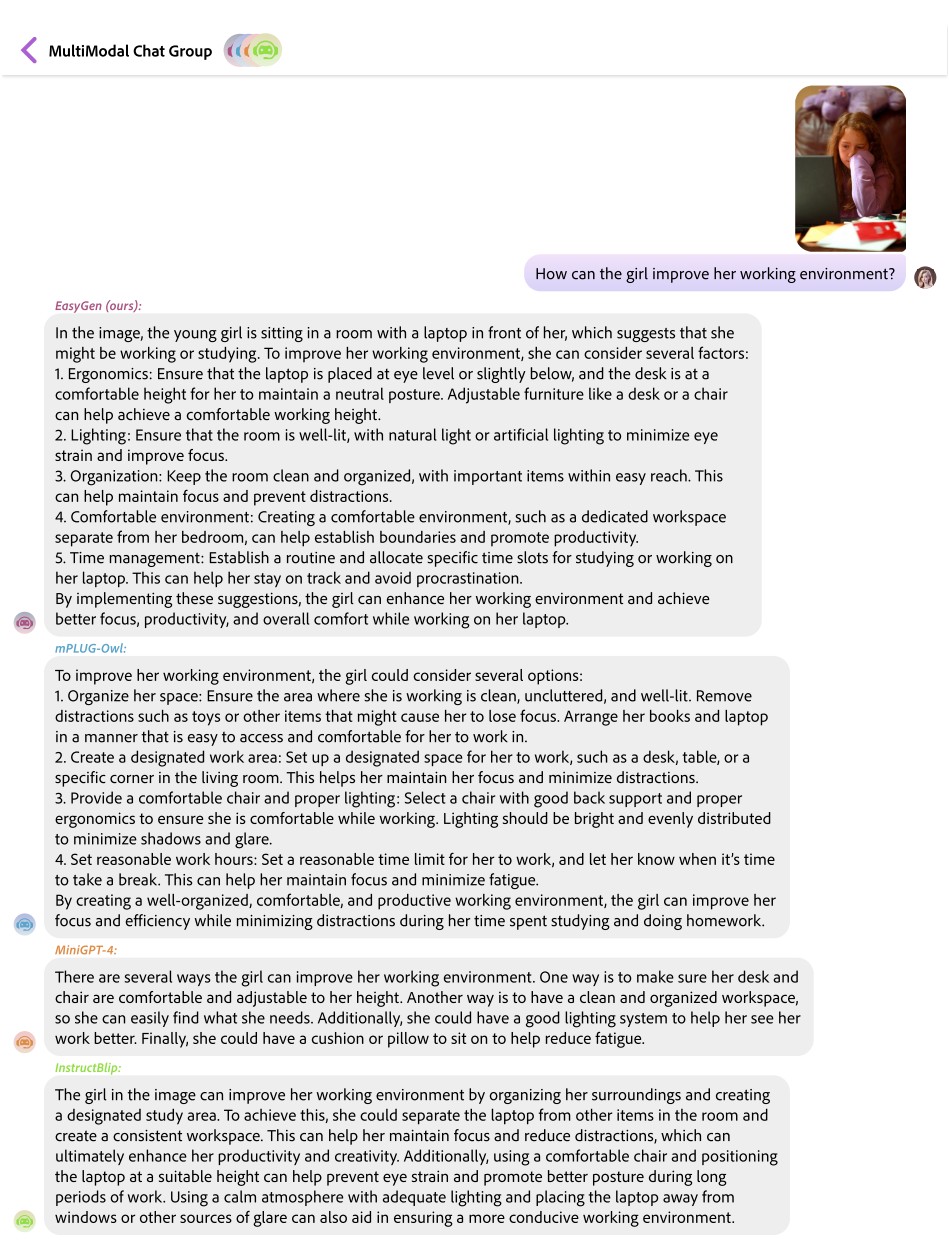

Figure 11: From this example, we can find that the response from EasyGen is more comprehensive and coherent. This shows EasyGen can give reasonable suggestions based on the given image.

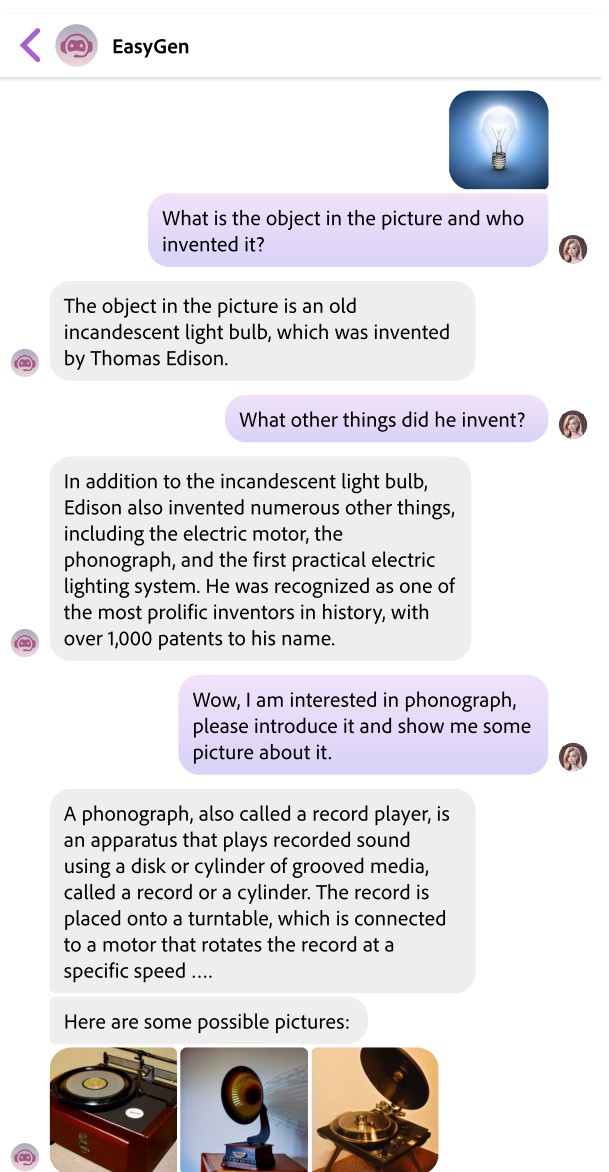

Figure 12: Example of multimodal response generation.

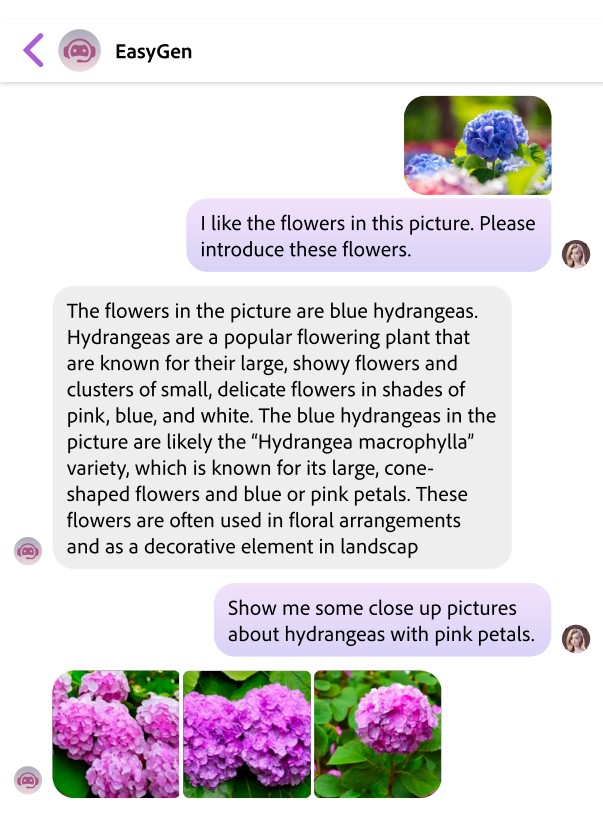

Figure 13: Example of multimodal response generation.

