# OpenReview forum: "Making Multimodal Generation Easier: When Diffusion Models Meet LLMS"
_ICLR.cc/2024/Conference — ICLR 2024 Conference Withdrawn Submission_

### Official Review · Reviewer_4W6z · 2023-10-23

**Soundness:** 3 good
**Presentation:** 2 fair
**Contribution:** 2 fair
**Rating:** 5
**Confidence:** 4

**Summary:**

The paper presents "EasyGen," a new multimodal model that enhances understanding and generation by utilizing diffusion models and large language models (LLMs). Unlike traditional models that rely heavily on encoders and significant training data, EasyGen employs a unique bidirectional diffusion model, "BiDiffuser." This allows for efficient image-to-text and text-to-image generation. Experimental results validate EasyGen's effectiveness, and its design is optimized for lab-based training.

**Strengths:**

This paper makes an early try on combining diffusion model and LLM to solve various downstream tasks.

The proposed method, EasyGen, can simultaneously generate image and text with much less training efforts than existing multimodal models.

The paper is easy to follow and experimental results are good on several benchmarks.

**Weaknesses:**

The novelty is limited. The proposed EasyGen simply marries the LLM with the UnifDiffuser. Meanwhile, in order to perform the text generation and image generation tasks, LLM should be plugged into different positions and retrained, which is redundant and loses the conciseness of large models.

It seems that the claimed “data efficient training” property of EasyGen comes from its UniDiffuser component, which can be regarded as a well-established multimodal interactor. Therefore, it is unfair to compare the data efficiency of EasyGen with other MLLM like BLIP, MiniGPT-4 and LLaVA (Table 4), which all start training from raw LLM and vision backbone.

A main weakness of existing diffusion models lies in that they cannot effectively comprehend long textual context due to its weak text encoder. Therefore, it is prospective to combine the LLM and diffusion models for stronger generation ability. Does the EasyGen possess such the ability?

**Questions:**

See the weakness part.

---

> ### Author Response · Authors · 2023-11-23
>
> Thanks for your comments!
>
> **1. The proposed EasyGen simply marries the LLM with the UnifDiffuser. Meanwhile, in order to perform the text generation and image generation tasks, LLM should be plugged into different positions and retrained, which is redundant and loses the conciseness of large models.**
>
> In order to make LLMs have the ability to understand image, current MLLMs such as LLaVA and MiniGPT-4 usually leverage instruction tuning method to fine-tune large models. This method can enhance the capability of multimodal interaction while maintaining the LLM capability.
> So our primary task is to employ an instruction tuning method to enable the model to generate image captions. Suitable instruction datasets could include PhotoChat, TextBind, and others.
>
> **2. It seems that the claimed “data efficient training” property of EasyGen comes from its UniDiffuser component, which can be regarded as a well-established multimodal interactor. Therefore, it is unfair to compare the data efficiency of EasyGen with other MLLM like BLIP, MiniGPT-4 and LLaVA (Table 4), which all start training from raw LLM and vision backbone.**
>
> We acknowledge the comparision is unfair. So we do the following experiments.
> To examine the efficiency of BiDiffuser, we combine its output with the image encoded by image CLIP ViT-L/14 and fine-tune the parameters of the LLM and projection layers. Also, w/o BiDiffuser means we directly align LLM with CLIP and using the same dataset from EasyGen. We fine-tune EasyGen using training and validation splits from VQAv2, Text Captions, AOK-VQA, and TextVQA datasets. The table below showcases the results of our method compared to leading models. To validate the effectiveness of the BiDiffuser module, we also incorporate it into LLaVA Vicuna-7B and use the same mixed datasets for instruction tuning. Given that BiDiffuser can convert images into text vectors, it can be directly transferred from EasyGen Vicuna-7B to the LLaVA Vicuna-7B model. In EasyGen, the model's performance significantly improves when using the same alignment and instruction tuning data with the BiDiffuser module.
>
> | Model | VQAv2 (test-dev) | TextVQA | MMbench (dev) |
> | -------- | -------- | -------- | -------- |
> | MiniGPT4 | - | 19.4 |   24.3   |
> | InstructBlip Vicuna-7B |  -   | 50.1 |  36.0  |
> | LLaVA Vicuna-7B | 77.6 | 44.1 |   43.6   |
> | LLaVA Vicuna-7B+BiDiffuser | 78.2 | 46.7 |   45.7   |
> | EasyGen ViT-L Vicuna-7B  | 79.4 | 45.5 |   45.4   |
> | EasyGen ViT-L Vicuna-7B (w/o BiDiffuser)   | 71.1 | 36.2 |   21.4   |
>
> **3. A main weakness of existing diffusion models lies in that they cannot effectively comprehend long textual context due to its weak text encoder. Therefore, it is prospective to combine the LLM and diffusion models for stronger generation ability. Does the EasyGen possess such the ability?**
>
> Since EasyGen is solely trained to produce image captions that can be comprehended by the diffusion model, we do not modify the diffusion model's text encoder. Integrating LLM as a text encoder for the diffusion model can enhance the alignment of the two models. However, this necessitates retraining the diffusion model, which is a resource-intensive process.

---

### Official Review · Reviewer_Sp3b · 2023-10-31

**Soundness:** 2 fair
**Presentation:** 3 good
**Contribution:** 3 good
**Rating:** 5
**Confidence:** 5

**Summary:**

This paper presents EasyGen, a method that fine-tunes a UniDiffuser [A] encoder with image-text bidirectional generation to enable LLMs to generate and understand multimodality grounded by diffusion models. The images are generated by creating textural inputs with Divter [B] based on LLMs, and textural responses are generated by first converting images to texts with the fine-tuned UniDiffuser and then reasoned by LLMs. Experiments on some classical VL tasks, including image captioning, short VQA tasks like GQA, and multimodal dialogue tasks on PhotoChat, demonstrate the promising performance of the proposed EasyGen.

[A] UniDiffuser: One Transformer Fits All Distributions in Multi-Modal Diffusion at Scale. In ICML 2023.\
[B] Multimodal Dialogue Response Generation. In ACL 2022.

**Strengths:**

- The idea of tuning UniDiffuser as LLM encoders via image-text bidirectional generation is simple and novel, and the results are interesting to me.
- The targeted problem of multimodal dialogue is relevant and trending, especially in the modern LLM era.
- The method seems very data-efficient, which is good.

**Weaknesses:**

My primary concern is the limited applications and evaluations. For multimodal dialogue, when considering long context conditional image generation or image conditional (multi-images or single-image) image generation like image edition/translation (it is, however, what UniDiffuser initially can do), only generating text inputs as diffusion modal conditions is limited.

For the evaluations, the current results only include traditional short QA or image captioning tasks. However, one of the advantages of LLMs is that they are mighty at world knowledge and long-context reasoning. It would be better if results on modern benchmarks like MM-Vet and MMBench were provided. Also, the most commonly used VQAv2 results are not provided.

On the other hand, since the method converts images to texts to obtain multimodal understandings, lots of information may be diluted or missing. It will harm the understanding that requires detailed knowledge, such as OCR-related tasks (results on TextVQA are also encouraged).  Besides, results on commonly used text-to-image generation benchmarks are required, such as COCO FID results.

It is somewhat overclaimed to criticize the large data requirement of CLIP/ImageBind models since UniDiffuser is also trained on large-scale corpora using large amounts of resources that cannot be affordable by most labs.

Missing citations and discussions. The method of first converting images to texts and then feeding them to LLMs is the same as Img2LLM [A], but it is not cited or discussed. Besides, several closely related concurrent works should be discussed [B-F].

[A] From Images to Textual Prompts: Zero-shot VQA with Frozen Large Language Models.
[B] DreamLLM: Synergistic Multimodal Comprehension and Creation.
[C] NExT-GPT: Any-to-Any Multimodal LLM.
[D] MiniGPT-5: Interleaved Vision-and-Language Generation via Generative Vokens.
[E] Generating Images with Multimodal Language Models.
[F] Kosmos-G: Generating Images in Context with Multimodal Large Language Models.

**Questions:**

- What about COCO captioning results on `test` split?
- Is is possible if some language capability can be provided? Such as MMLU and HellaSwag benchmarks.

---

> ### Author Response · Authors · 2023-11-23
>
> Thanks for your comments!
>
> **1. For multimodal dialogue, when considering long context conditional image generation or image conditional (multi-images or single-image) image generation like image edition/translation (it is, however, what UniDiffuser initially can do), only generating text inputs as diffusion modal conditions is limited. For the evaluations, the current results only include traditional short QA or image captioning tasks. However, one of the advantages of LLMs is that they are mighty at world knowledge and long-context reasoning. It would be better if results on modern benchmarks like MM-Vet and MMBench were provided. Also, the most commonly used VQAv2 results are not provided.**
>
> We acknowledge that converting images to text may result in loss of information, hence we only apply EasyGen for image-captioning and zero-shot VQA tasks. To fully utilize the image's information while considering the significant cost of fine-tuning the diffusion model for the VQA task, we choose to combine the output of BiDiffuser with the image encoded by image CLIP ViT-L/14. We then fine-tune the parameters of the LLM and projection layers. EasyGen is fine-tuned using training and validation splits from VQAv2, Text Captions, AOK-VQA, and TextVQA datasets. The table below illustrates the results of our method compared to leading models. To confirm the effectiveness of the BiDiffuser module, we also incorporate it into LLaVA Vicuna-7B and use the same mixed datasets for instruction tuning. Given that BiDiffuser can convert images into text vectors, it can be directly transferred from EasyGen Vicuna-7B to the LLaVA Vicuna-7B model. The results indicate that BiDiffuser aids the model in understanding images.
>
>
> | Model | VQAv2 (test-dev) | TextVQA | MMbench (dev) |
> | -------- | -------- | -------- | -------- |
> | MiniGPT4 | - | 19.4 |   24.3   |
> | InstructBlip Vicuna-7B |  -   | 50.1 |  36.0  |
> | LLaVA Vicuna-7B | 77.6 | 44.1 |   43.6   |
> | LLaVA Vicuna-7B+BiDiffuser | 78.2 | 46.7 |   45.7   |
> | EasyGen ViT-L Vicuna-7B  | 79.4 | 45.5 |   45.4   |
> | EasyGen ViT-L Vicuna-7B (w/o BiDiffuser)   | 71.1 | 36.2 |   21.4   |
>
> **2. What about COCO captioning results on test split?**
>
> In our study, we utilize the COCO Karpathy test split as the COCO captioning test split.
>
> **3. It is somewhat overclaimed to criticize the large data requirement of CLIP/ImageBind models since UniDiffuser is also trained on large-scale corpora using large amounts of resources that cannot be affordable by most labs.**
>
> We acknowledge the comparision is unfair. So we do the experiments. However, when multimodal language models use them as a visual module, CLIP/ImageBind still requires large-scale corpora to achieve alignment. In our experiment, the volume of data needed for fine-tuning and aligning the UniDiffuser process is significantly reduced.
>
> **4. Missing citations and discussions. The method of first converting images to texts and then feeding them to LLMs is the same as Img2LLM, but it is not cited or discussed. Besides, several closely related concurrent works should be discussed.**
>
> Thanks for your advice, we will add the missing citations and discussions to our related works in the revision

---

> > ### Comment · Reviewer_Sp3b · 2023-11-23
> > **Why it is zero-shot?**
> >
> > You have used training and validation data for fine-tuning. Why it is zero-shot?
> >
> > Besides, ICLR allows authors to revise the paper during rebuttal.

---

> > > ### Author Response · Authors · 2023-11-23
> > >
> > > There may have been some confusion. Our zero-shot evaluation specifically pertains to the results presented in Table 1. However, after conducting fine-tuning for the VQA task, we have obtained updated results, which are now reflected in Table 5. We have made revisions to the manuscript accordingly, and the newly added content is highlighted in blue font. Please review the updated version.

---

### Official Review · Reviewer_4jWn · 2023-10-31

**Soundness:** 3 good
**Presentation:** 3 good
**Contribution:** 2 fair
**Rating:** 5
**Confidence:** 4

**Summary:**

This paper unveils EasyGen, a model purposed for enhancing multimodal understanding and generation by synergizing diffusion models with large language models (LLMs). The central innovation, EasyGen, employs a Bi-directional conditional diffusion model, dubbed BiDiffuser, which connects diverse modalities, thereby smoothing the path for both image-to-text and text-to-image generation. Demonstrated across an array of tasks such as image captioning, visual question answering, and multimodal dialogue generation, EasyGen displays commendable performance, establishing its competitive stance.

**Strengths:**

(1) Exhibiting a novel fusion of diffusion models with LLMs for multimodal generation.

(2) With the advent of the BiDiffuser model, a consequential bridge is established between image and text modalities, thereby catalyzing enhanced interactivity amongst them.

(3) In the realm of various multimodal tasks, EasyGen asserts its dominance by showcasing competitive performance, maintaining commendable results despite operating on a comparatively lesser volume of training data.

**Weaknesses:**

(1) While the paper ardently seeks to amalgamate two predominant techniques, LLM and diffusion, the introduction of the BiDiffuser appears somewhat diminished in novelty due to its considerable reliance on the pre-existing UniDiffuser. It presents itself more as a refined version of the latter, which in turn slightly hampers the overall technical contribution.

(2) Elements such as pre-alignment and mid-alignment do not introduce fresh perspectives; instead, they lean on existing paradigms. The proposed ITG and ITM losses, while operational, seem somewhat pedestrian in their approach which would be hard to converge to zero, casting doubts over their capacity to effectively aligning the LLM and diffusion models.

(3) The claim of data efficiency, a major highlighted merit, appears somewhat tenuous, given the possibility of the BiDiffuser being a fine-tuned iteration of a pre-trained UniDiffuser.

(4) The paper could further benefit from more details of instruction-tuning. Incorporating concrete examples instead of mere templates in Section 3.2.2 would be much helpful for comprehension.

(5) For a more streamlined reader experience, it would be advantageous to highlight the foremost results in Tables 4 and 6.

(6) Consider revising "RELATED WORK" to "RELATED WORKS" for terminological consistency.

**Questions:**

N/A

---

> ### Author Response · Authors · 2023-11-23
>
> Thanks for your comments!
>
> **1. The claim of data efficiency, a major highlighted merit, appears somewhat tenuous, given the possibility of the BiDiffuser being a fine-tuned iteration of a pre-trained UniDiffuser.**
>
> We recognize that the comparison might not be completely fair. So we do the following experiments.
> To evaluate the effectiveness of BiDiffuser, we incorporate its output with the image encoded by image CLIP ViT-L/14 and fine-tune the parameters of the LLM and projection layers. We conduct fine-tuning on the training and validation splits of VQAv2, Text Captions, AOK-VQA, and TextVQA datasets. The table presented here showcases the results of our approach compared to state-of-the-art models. Additionally, we integrate the BiDiffuser module into LLaVA Vicuna-7B and perform instruction tuning using the same mixed datasets. Since BiDiffuser can map images to text vectors, the BiDiffuser in EasyGen Vicuna-7B can be directly transferred to the LLaVA Vicuna-7B model. In EasyGen, with the same alignment and instruction tuning data, the model's performance significantly improves when the BiDiffuser module is incorporated.
>
> | Model | VQAv2 (test-dev) | TextVQA | MMbench (dev) |
> | -------- | -------- | -------- | -------- |
> | MiniGPT4 | - | 19.4 |   24.3   |
> | InstructBlip Vicuna-7B |  -   | 50.1 |  36.0  |
> | LLaVA Vicuna-7B | 77.6 | 44.1 |   43.6   |
> | LLaVA Vicuna-7B+BiDiffuser | 78.2 | 46.7 |   45.7   |
> | EasyGen ViT-L Vicuna-7B  | 79.4 | 45.5 |   45.4   |
> | EasyGen ViT-L Vicuna-7B (w/o BiDiffuser)   | 71.1 | 36.2 |   21.4   |
>
> **2. The paper could further benefit from more details of instruction-tuning. Incorporating concrete examples instead of mere templates in Section 3.2.2 would be much helpful for comprehension. For a more streamlined reader experience, it would be advantageous to highlight the foremost results in Tables 4 and 6.**
>
> Thanks for you advice! We will add concrete examples Section 3.2.2 and highlight the foremost results in Tables 4 and 6. Also revising "RELATED WORK" to "RELATED WORKS" .

---

### Official Review · Reviewer_E4up · 2023-11-04

**Soundness:** 2 fair
**Presentation:** 2 fair
**Contribution:** 2 fair
**Rating:** 3
**Confidence:** 4

**Summary:**

This study aims to build a framework for multimodal generation for image-to-text and text-to-image. After fine-tuning UniDiffusion, which jointly learns the distribution of texts and images, into BiDiffuser, the authors connect the BiDiffuser with small Language Models such as FlanT2XL with 3B parameters or Vicuna-7B. Meanwhile, the proposed framework requires to separately train a model for text-to-image and image-to-text, respectively. For the task of image-to-text, two aligning methods, pre-align and mid-align manners, are proposed. The experimental results show that the proposed framework can outperform previous frameworks despite smaller training dataset for the alignment.

**Strengths:**

S1. This study covers a timely research topic to expand LMs into multimodal generative tasks including instruction-based text-to-image and image-to-text.

S2. An intuitive and effective alignment methods, pre-align and mid-align, are proposed for decoder-only LMs and encoder-decoder LMs, respectively.

S3. The proposed framework, EasyGen, reports remarkable performance on benchmark datasets, NoCaps, COCO, OK-VQA, and GQA.

**Weaknesses:**

W1. The proposed framework, EasyGen, has a limited flexibility to conduct multimodal generative tasks. EasyGen requires to separately train a framework for text-to-image and image-to-text.

W2. The proposed framework does not solely depend on aligning the representations of diffusion models and LMs, but fine-tunes LMs to improve the performance.

W3. The organization and presentation of this paper should be improved. For example, the formal definition of the task of instruction-based multimodal generation lacks, although the proposed framework aims to conduct the task. This paper merely presents the proposed methods in parallel. In addition, Section 3.2.2 is introduced after Section 3.1.1, while Section 3.2.2 is necessary to understand Section 3.2.1 and Section 3.3. Despite UniDiffusion is the core model architecture of BiDiffuser, this paper assumes that the readers already know the details of UniDiffusion and does not explain them, such as how the diffusion forward process of images and texts are defined and how the embedding of texts and images are determined.

W4. Despite remarkable results on benchmarks of image-to-text including image captioning and VQA, the core contributions should be demonstrated by thorough experiments. Please refer to the questions below.

W5. Section 5, Related Work, should be improved. Especially, in the paragraph of “Multimodal Language Models,” the authors merely list-up previous model names without proper explanations and comparisons with the proposed approach.

**Questions:**

Q1. In Section 3.1, I wonder whether each modality uses the same forward process or different forward processes to corrupt the representations of images and texts. In addition, how the texts and images are encoded into the common representation space?

Q2. To clearly understand Figure 4, I wonder whether BiDiffuser extracts the features of textual noise by using only one inference or using iterative denoising. If it uses iterative denoising of diffusion models, the encoding time of image representation and the generation speed of image-to-text should be compared with other approaches.

Q3. For the tasks of text-to-image and image-to-text, respectively, does the finetuning of LMs update the entire parameters of LMs? Can LoRA or prompt tuning be applied for efficient fine-tuning of LMs?

Q4. This study only uses small LMs with less than 10B parameters, such as 3B or 7B parameters. How about the LMs with larger parameters, such as LLaMA-13B or LLaMA-70B, are used? Can enlarging the size of LMs more improve the performance of EasyGen under the same training dataset?

Q5. For the task of image-to-text, <query> is located after <image>. Is the ordering of <image> and <query> important?

Q6. Why is the performance of EasyGeN with Vicuna-7B on OK-VQA much inferior to LLaVA, while both EasyGen and LLaVA use the Vicuna model? If the dataset size increases, can EasyGen outperform LLaVA? In Section 1, the authors claim that combining a bidirectional conditional diffusion model with language models can decrease the gap between the two. If the proposed framework also requires a similar size of dataset with LLaVA, is the authors’ claim still valid?

Q7. How much total dataset of multimodal samples (image-text) are used to train LMs, UniDiffusion, BiDiffuser, and the alignment modules, compared with other approaches?

Q8. Why does Table 7 exclude the results of GQA? Do the results show different trends compared with other tasks? In addition, I wonder about the performance of (freezed Vicuna-7B + UniDiffuser + Pre-Align), since the table shows that BiDiffuser is the key to improve the performance rather than the alignment methods. It is also interesting to see the performance of (freezed T5, UniDiffuser, Mid-Align) to further examine the effectiveness of BiDiffuser.

Q9. Since measuring the performance of generative models can be biased on benchmarks, I suggest the authors conduct a user study to compare with other methods such as BLIP-2 and LLaVA.

Q10. How can a bidirectional conditional diffusion model reduce the gap between LMs? Is the diffusion model mandatory to reduce the gap? Which property of diffusion models, which iteratively denoise latents to generate texts, affect to reduce the gap?

Q11. In the image-to-text pipeline, how about using the embeddings of image for the projection layer instead of the textual noise? Since this framework uses the features of texts, I think that the projection layer aligns the text features of BiDiffuser with LMs instead of the image features with LMs. Thus, the difference between using BiDiffuser and any image-captioning model is ambiguous.

Q12. For the task of text-to-image, why should the LLM be fine-tuned? I think that it’s not an alignment approach between LLM and image decoder (BiDiffuser), but merely fine-tuning LLMs as a tailored model to text-to-image.

Q13. Why is the FID score of EasyGen w/o generated desc in Table 5 abnormally high? How about the qualitative results of generated images? What is the reason for the abnormally high FID?

Q14. The authors have fine-tuned UniDiffuer on MS-COCO for text-to-image and image-to-text. What if the BiDiffuser is finetuned on all the tasks in UniDiffuser, while learning MS-COCO dataset? In addition, what if the training dataset is changed into other datasets, collected in the wild setting such as LAION or COYO? I guess that the remarkable performance of the proposed framework might come from the MS-COCO dataset, since the distribution of (image, text) is similar with benchmarks such as (NoCaps, COCO, OK-VQA, GQA) in Table 3.

---

> ### Author Response · Authors · 2023-11-23
>
> Thanks for your comments!
>
> 1. Each modality employs the same forward process to corrupt its representations, following the same schedule. The distinction lies in the addition of 4X64X64 Gaussian noise to the image (encoded by AutoencoderKL) and 77X768 Gaussian noise to the text (encoded by CLIP text encoder). As UniDiffuser can integrate different modalities, we can use the text representation as input to the LLM to aid in image comprehension.
>
> 2. BiDiffuser utilizes an iterative denoising method, specifically employing DPM-Solver for sampling with a step number of 50. In our experiments, we tested our model's performance on an A100(80G). We used 1000 image-caption pairs for inference. On average, EasyGen took 2.95 seconds for the caption generation task (with the diffusion module taking about 2.41 seconds), and approximately 4.96 seconds to generate an image.
>
> 3. In our tasks, we updated all the parameters of the Language Models (LMs). We also attempted efficient instruction-tuning using LoRA:
>
> | Model | VQAv2 (test-dev) | TextVQA | MMbench (dev) |
> | -------- | -------- | -------- | -------- |
> | MiniGPT4 | - | 19.4 |   24.3   |
> | InstructBlip Vicuna-7B |  -   | 50.1 |  36.0  |
> | LLaVA Vicuna-7B | 77.6 | 44.1 |   43.6   |
> | LLaVA Vicuna-7B+BiDiffuser | 78.2 | 46.7 |   45.7   |
> | EasyGen ViT-L Vicuna-7B  | 79.4 | 45.5 |   45.4   |
> | + w/o BiDiffuser   | 71.1 | 36.2 |   21.4   |
>
> 4. We are very sorry, as our experiments were conducted in a lab setting, we only studied our method on 7B LLMs.
>
> 5. Custom data. In line with current works such as LLaVA and InstructBLIP, we positioned <query\> after <image\>.
>
> 6. We believe the primary reason for the inferior performance of EasyGeN with Vicuna-7B on OK-VQA compared to LLaVA is the instruction tuning data. As demonstrated in the table, to verify the effectiveness of the BiDiffuser module, we also added this module to LLaVA Vicuna-7B and used the same mixed datasets for instruction tuning. Since BiDiffuser can convert images into text vectors, it can be directly transferred from EasyGen Vicuna-7B to the LLaVA Vicuna-7B model. The results indicate that BiDiffuser aids the model in understanding images. Both LLaVA and our model use the same instruction tuning datasets. It's worth noting that LLaVA has used 595K pretraining data from the CC-3M dataset to align CLIP and LLM. Our model does not pre-align CLIP and LLM, and only uses instruction-tuning data for training.
>
> 7. In our alignment and VQA fine-tuning stages, this table provides a description of the datasets we utilized. It is important to mention that during the alignment process, we employed 5,000 images from the VQAv2 dataset and an additional 5,000 images from the VisDial dataset.
>
> | Data types | Dataset | Size | BiDiffuser | Alignment | Fine-tuning |
> | -------- | -------- | -------- |-------- |-------- |-------- |
> | Caption     | MS-COCO caption | 83K | Y | Y | N |
> | Caption     | Visual Genome | 83K | Y | N | N |
> | Multimodal instruction  | LLaVA dataset | 80K | N | Y | Y |
> | VQA  | VQAv2 | 83K | N | - | Y |
> | VQA  | AOK-VQA | 66K | N | N | Y |
> | OCR-related tasks  | Text Captions | 22K | N | N | Y |
> | OCR-related tasks  | TextVQA | 22K | N | N | Y |
>
> 8. Since Table 7 displays the results of the ablation study and OK-VQA has a smaller size compared to the GQA test split, we only evaluate these models on the OK-VQA test split.
>
> 9. To compare our models with other methods in the benchmarks, we included MMbench. The results can be found in the table above.
>
> 10. The bidirectional conditional diffusion model can narrow the gap between language models (LMs) by utilizing text embeddings to represent images. While diffusion model is not necessary for reducing the gap, BLIP2, for example, utilizes Q-Former to enable learnable inputs to capture image information. However, the bidirectional conditional diffusion model can also learn to generate images.
>
> 11. Unlike CLIP, our model leverage BiDiffuser's text embedding as image's representation and our projection layer aligns the text features of BiDiffuser with LMs, making it easy to train the projection layer and achieve alignment. Just using the embedding of image usually need a large amount of image-text pairs in the alignment stage. Additionally, we took into consideration that converting images to texts may result in missing information. Therefore, we chose to concatenate the output of BiDiffuser with the image encoded by image CLIP ViT-L/14 and fine-tune the parameters of the LLM and projection layers. The experiments' results can be found in Q3's table.

---

> > ### Author Response · Authors · 2023-11-23
> >
> > 10. The bidirectional conditional diffusion model can narrow the gap between language models (LMs) by utilizing text embeddings to represent images. While diffusion model is not necessary for reducing the gap, BLIP2, for example, utilizes Q-Former to enable learnable inputs to capture image information. However, the bidirectional conditional diffusion model can also learn to generate images.
> >
> > 11. Unlike CLIP, our model leverage BiDiffuser's text embedding as image's representation and our projection layer aligns the text features of BiDiffuser with LMs, making it easy to train the projection layer and achieve alignment. Just using the embedding of image usually need a large amount of image-text pairs in the alignment stage. Additionally, we took into consideration that converting images to texts may result in missing information. Therefore, we chose to concatenate the output of BiDiffuser with the image encoded by image CLIP ViT-L/14 and fine-tune the parameters of the LLM and projection layers. The experiments' results can be found in Q3's table.
> >
> > 12. We fine-tuned the LLM to generate captions that can be understood by the diffusion model. This explains why the FID score of EasyGen w/o generated desc in Table 5 is unusually high. As a lower FID indicates better quality of generated images. That means the captions generated by EasyGen w/o generated desc are not suitable for BiDiffuser. The image descriptions in the PhotoChat dataset are too brief to adequately convey the image information. The diffusion model struggles to comprehend the concise context due to its weak text encoder. Hence, we utilized the pre-trained model from the first stage to regenerate the image descriptions.
> >
> > 13. Since our experiments were conducted in a controlled laboratory environment, it is challenging to fine-tune BiDiffuser on all the tasks in UniDiffuser. In our comparative baselines, their training sets also include the MS-COCO dataset. So we think our model can benefit from BiDiffuser.